

# Dietary diversity is inversely correlated with pre-pregnancy body mass index among women in a Michigan pregnancy cohort

Breanna M. Kornatowski and Sarah S. Comstock

Department of Food Science and Human Nutrition, Michigan State University, East Lansing, MI, United States of America

## ABSTRACT

**Background**. During pregnancy, healthy maternal body weight and a nutritionally complete diet provide a favorable environment for fetal development. Yet nearly two-thirds of women of reproductive age in the United States (US) are either overweight or obese. The objective of this research was to investigate the relationship between a measure of dietary diversity and body mass index (BMI) in a sample of women enrolled in a US pregnancy cohort.

**Methods**. Dietary data was obtained from one 24-hour dietary recall collected during the third trimester of pregnancy ($n = 40$). Pre-pregnancy BMI was calculated from pre-pregnancy weight and height self-reported by survey at the time of enrollment. Using the Minimum Dietary Diversity for Women (MDD-W) indicator developed by the Food and Agriculture Organization of the United Nations, dietary data was categorized and scored.

**Results**. Overall, 35% of participants did not achieve minimum dietary diversity (MDD-W $\geq$ 5). In addition, 45% of participants were obese, 35% were normal weight and 20% were overweight. Women with higher BMI had lower MDD-W scores than women with lower BMI ($p < 0.05$). The median MDD-W for both normal and overweight women was $\geq$ 5 indicating that normal and overweight pregnant women tended to consume a diet that was high in micronutrients. Conversely, the median MDD-W for obese women was below 5 meaning these women tended to consume a diet that was low in micronutrient density. The most commonly consumed food group was grain. In addition, 75% of all participants consumed sweetened drinks. Individuals with an MDD-W score $\geq$ 5, were more likely to have consumed dark green leafy vegetables, vitamin A-rich fruits or vegetables, other vegetables and other fruits than those with MDD-W scores <5.

**Discussion**. In this study, we show that a food group diversity indicator that has been shown to reflect adequacy of micronutrient intake in populations from less economically developed countries may also be informative in US populations. Furthermore, these results reflect the importance of encouraging all pregnant women with less varied diets to consume more fruits and vegetables.

Corresponding author
Sarah S. Comstock,
comsto37@msu.edu

# INTRODUCTION

One in four American women are obese prior to becoming pregnant (*Branum, Kirmeyer & Gregory, 2016*). Obesity is associated with numerous health issues (*Williams et al., 2015*), especially in pregnancy. In fact, an unhealthy body mass index (BMI) and diet in pregnancy exerts negative effects on the developing fetus and leads to health problems for the child later in life (*Godfrey et al., 2017*; *Poston et al., 2016*). Furthermore, women who are obese are more likely to have children who are obese (*Oestreich & Moley, 2017*), perpetuating health problems from generation to generation. There is evidence that obese individuals are deficient in micronutrients (*Costa et al., 2018*; *Krzizek et al., 2018*; *Stankowiak-Kulpa et al., 2017*).

Women of reproductive age require a more nutrient-dense diet because of the physiological needs during pregnancy (*Gernand et al., 2016*). Commonly, women of reproductive age consume supplemental micronutrients including iron, folic acid, and zinc (*Gernand et al., 2016*). An insufficient amount of micronutrients during pregnancy can create complications such as congenital malformations in the baby and increased risk of hemorrhage during delivery (*Black, 2001*). In addition, insufficient micronutrients can inhibit physical, neurobehavioral, and immunological growth among other complications (*Gernand et al., 2016*; *Ojha et al., 2013*; *Sullivan et al., 2009*; *Zerfu & Ayele, 2013*). Consuming prenatal vitamins can provide these micronutrients, however, the "most desirable" way to meet micronutrient needs is through a "sustained diet of various micronutrient-dense foods" (*Gernand et al., 2016*; *Hanson et al., 2015*). Unfortunately, only around 78 percent of pregnant women in the United States recently reported taking some sort of multivitamin while they were pregnant, making it even more important for women to consume micronutrients through their diet (*Sullivan et al., 2009*).

Although it is known that micronutrient intake is important in pregnancy and that obesity in pregnancy can lead to negative birth outcomes, the relationship between pre-pregnancy BMI and micronutrient intake has been inadequately assessed. One excellent way to determine micronutrient status of a population is to apply the Food and Agriculture Organization's (FAO) dietary diversity score (*FAO & FHI 360, 2016*). Dietary diversity is not only a predictor for micronutrient status and diet quality, but it is also a predictor of obesity. In a study done by *Wolongevicz et al. (2010)* women with lower diet quality were significantly more likely to be overweight or obese compared to women with higher diet quality.

Despite the importance of dietary intake during pregnancy, due to effects of diet on pregnancy outcomes and child health, little research has examined the potential association between pre-pregnancy BMI and dietary diversity in pregnancy. A diet of high micronutrient adequacy can be achieved through a diverse diet that includes many different food groups. The minimum dietary diversity for women (MDD-W) indicator assesses micronutrient adequacy by determining whether a subject consumed at least five out of ten categories of food groups in the past day (*Martin-Prevel et al., 2015*). A score of ≥5 indicates the individual has a diet of an estimated micronutrient adequacy and has a diverse diet. In a given population, the proportion of women who achieve dietary diversity

can be used as an indicator for higher micronutrient adequacy (*FAO & FHI 360, 2016*). The objective for this study was to measure diversity in dietary intake by women during their third trimester of pregnancy and to determine if MDD-W score was associated with pre-pregnancy BMI. The hypothesis for this study was that obese women would have lower MDD-W scores than women of normal or healthy weight, meaning obese women have diets with a lower estimated micronutrient adequacy.

## MATERIALS AND METHODS

### Participants

This study used data from two studies, one study was nested in the Archive for Research on Child Health (ARCH) called ARCH GUT and the other study was a stand alone study called BABY GUT. The ARCH study is a cohort assembled in pregnancy to examine relationships between exposures assessed prospectively in pregnancy and the health and development of children. A total of 1,043 women were enrolled in the ARCH cohort from 2008–2017 with 53 women enrolled in ARCH during the time ARCH GUT was recruiting (Nov 2015–2017). ARCH exclusion criteria were age <18 years and inability to answer a questionnaire in English. ARCH women were recruited at their first prenatal visit (mean gestational age at enrollment, 13.4 weeks); were briefly interviewed about nutrition, physical activity, depression and socio-economic status; consented to the archiving of three urine specimens and extra blood obtained at two clinically-required phlebotomies; allowed access to birth certificates, medical records and the state-archived newborn blood spot; and consented to postnatal follow-up. BABY GUT recruitment relied on responses to flyers placed in several OB/GYN clinics in the Lansing, MI area. Exclusion criteria were as in ARCH, but also excluded women with BMI <18.5. Women could enroll at any point in pregnancy, but data for this study were collected during the third trimester. All participants provided written informed consent. Study procedures for ARCH (IRB#C07-1201), ARCH GUT (IRB# 14-170M) and BABY GUT (IRB# 15-1240) were approved by the Michigan State University IRB.

Data presented here were collected from 40 pregnant women who were enrolled in either ARCH GUT ($n = 25$) or BABY GUT ($n = 15$) in the Lansing and Traverse City areas of the state of Michigan, USA. All 40 participants completed self-administered questionnaires that included an open recall of dietary intake over a 24-hour period. Participants were asked to report age, pre-pregnancy weight, height, and parity. The validity of self-reported pre-pregnancy weight and height to determine pre-pregnancy weight status classification has been confirmed previously (*Shin et al., 2014*). Participant characteristics are shown in Table 1.

### Diet diversity analysis

The 24-hour dietary recall data was used to determine which of the 22 MDD-W categories (*FAO & FHI 360 , 2016*) a participant had consumed. These 22 categories included: grains; white roots and tubers, and plantains; pulses (beans, peas and lentils); nuts and seeds; milk and milk products; organ meat; meat and poultry; fish and seafood; eggs; dark green leafy vegetables; vitamin A-rich vegetables, roots and tubers; vitamin A-rich fruits; other
**Table 1  Participant characteristics by study.[a]**

| Individual characteristics | Overall | BABY GUT | ARCHGUT | *p*-value |
|---|---|---|---|---|
| *n* (% of Sample) | 40(100) | 15(37.5) | 25(62.5) | |
| **BMI** (*p* = 0.1395) | | | | |
| Normal | 14(35) | 6(40) | 8(32) | *0.7356* |
| Overweight | 8(20) | 5(33.3) | 3(12) | *0.1256* |
| Obese | 18(45) | 4(26.7) | 14(56) | *0.1040* |
| **Age Group** (*p* = 0.5766) | | | | |
| 20–24 | 2(5) | 1(6.7) | 1(4) | *1.00* |
| 25–29 | 10(25) | 2(13.3) | 8(32) | *0.2686* |
| 30–34 | 14(35) | 6(40) | 8(32) | *0.2333* |
| 35–39 | 14(35) | 6(40) | 8(32) | *0.2333* |
| **Parity** (*p* = 0.1030) | | | | |
| 1 | 14(35) | 5(33.3) | 9(36) | *1.00* |
| 2 | 20(50) | 10(66.7) | 10(40) | *0.1908* |
| 3+ | 6(15) | 0 | 6(24) | *0.0670* |

**Notes.**

[a]Comparisons across categories (i.e., normal, overweight, obese) within a characteristic (i.e., BMI) were compared between study cohorts using Fisher's exact test with *p* values reported next to characteristic names. Proportions within each category (i.e., normal) of each characteristic were compared between study cohorts using Fisher's exact test with *p*-values reported in the far right column.

vegetables; other fruits; insects and other small protein foods; red palm oil; other oils and fats; savory and fried snacks; sweets; sugar-sweetened beverages; condiments and seasonings; other beverages and foods. A subset (10 groups, Table 2) of the 22 categories were used to compute the MDD-W as described (*FAO & FHI 360 , 2016*). The MDD-W is a dichotomous scale with a cut-off based on consumption of at least five of the following 10 food groups: (1) Grains, roots, and tubers; (2) Pulses; (3) Nuts and seeds; (4) Dairy; (5) Meat, poultry, and fish; (6) Eggs; (7) Dark leafy green vegetables; (8) Other Vitamin-A rich fruits and vegetables; (9) Other vegetables; and (10) Other fruits. A subset of three groups (savory and fried snacks, sweets, sugar-sweetened beverages) was used to assess intake of low-nutrient-density foods. Briefly, if a woman consumed one or more items in a category, she received a point for that category. For example, if a woman consumed cheese (dairy), milk (dairy), eggs (eggs), and spinach (dark green leafy vegetables), she would receive one point for dairy, one point for eggs and one point for dark green leafy vegetables resulting in a final score of 3. Two researchers independently scored dietary intake data, and there were no disagreements in scoring results. No registered dietitians were involved in this project. Participants with scores of 5 or above were considered to have achieved minimum dietary diversity, based on the dichotomous scoring rubric (*FAO & FHI 360 , 2016*). Women were assigned a BMI category using the United States Centers for Disease Control definitions (*Garrow & Webster, 1985*; *NIH, 1998*): normal or healthy weight (18.5 ≤ BMI <25), overweight (25 ≤ BMI <30), obese (BMI ≥ 30).

## Statistical analyses

Fisher's exact test was used to compare ARCH GUT and BABY GUT participant characteristics as well as to compare among groupings within participant characteristics

**Table 2 MDD-W food group consumption according to BMI category and total MDD-W score.** Values indicate percent within BMI category who received the corresponding overall MDD-W score and consumed that particular category of food.

| | Normal weight | | Overweight | | Obese | | Overall | |
|---|---|---|---|---|---|---|---|---|
| Sample size (%) | 35 | | 20 | | 45 | | 100 | |
| Sample Size ($n$) | 2 | 12 | 1 | 7 | 11 | 7 | 14 | 26 |
| MDD-W Score | <5 | ≥ 5 | <5 | ≥ 5 | <5 | ≥ 5 | <5 | ≥ 5 |
| Grains | 100 | 100 | 100 | 100 | 90.9 | 100 | 92.9 | 100 |
| Pulses (brans, peas, and lentils) | 0 | 16.7 | 0 | 28.6 | 0 | 14.3 | 0 | 19.2 |
| Nuts and seeds | 0 | 41.7 | 0 | 42.9 | 9.1 | 28.6 | 7.1 | 38.5 |
| Dairy | 100 | 91.7 | 0 | 85.7 | 81.8 | 85.7 | 78.6 | 88.5 |
| Meat, fish, or poultry | 50 | 91.7 | 100 | 85.7 | 81.8 | 85.7 | 78.6 | 88.5 |
| Eggs | 0 | 33.3 | 100 | 28.6 | 9.1 | 42.9 | 14.3 | 34.6 |
| Dark green leafy vegetables | 50 | 50 | 0 | 42.9 | 0 | 57.1 | 7.1 | 50[a] |
| Other Vitamin A rich fruits or vegetables | 0 | 33.3 | 0 | 28.6 | 0 | 71.4 | 0 | 42.3[a] |
| Other vegetables | 50 | 75 | 0 | 57.1 | 27.3 | 57.1 | 28.6 | 65.4[a] |
| Other fruits | 0 | 75 | 0 | 100 | 9.1 | 71.4 | 7.1 | 80.8[a] |

Notes.

[a] indicates that individuals consuming fewer than five food groups were less likely to have consumed a particular food group than individuals consuming equal to or more than five food groups. Overall comparisons of intake between individuals consuming fewer than five food groups compared to those consuming equal to or more than five food groups were tested using Fisher's exact test.

(such as BMI, age, etc.) for all participants. Fisher's exact test was also used to compare the percent of individuals consuming a specific food category between those who achieved MDD-W scores of ≥5 versus those who did not. Spearman rank correlation was used to assess the association between total MDD-W score and BMI. Fisher's exact test was used to compare the percent of individuals achieving a MDD-W score ≥5 across the three BMI categories. Kruskal–Wallis was used to compare total MDD-W scores across BMI categories followed by pairwise comparisons using the Dwass, Steel, Critchlow-Fligner multiple comparison procedure. All statistical analyses were conducted with SAS version 9.4 (Cary, NC).

# RESULTS

## Participant characteristics

Participant characteristics did not differ between ARCH GUT and BABY GUT (Table 1). There was a trend for more ARCH GUT participants to be obese compared to BABY GUT participants ($p = 0.104$). Most of the women in both ARCH GUT and BABY GUT were between the ages of 30 and 39 years. Few women in either study had given birth to more than two children.

To calculate an MDD-W score for each participant, consumption of 10 food groups was assessed. Those pregnant women with MDD-W scores ≥5 more commonly consumed vegetables and fruits than did women with MDD-W scores <5 (Table 2). Overall, 35% of participants did not achieve minimum dietary diversity. All participants, with the exception of one, consumed grains. The majority of participants also consumed dairy. Pulses such as beans, peas, and lentils were consumed the least. None of the participants with low dietary diversity consumed pulses, and only 19.2% of participants achieving minimum
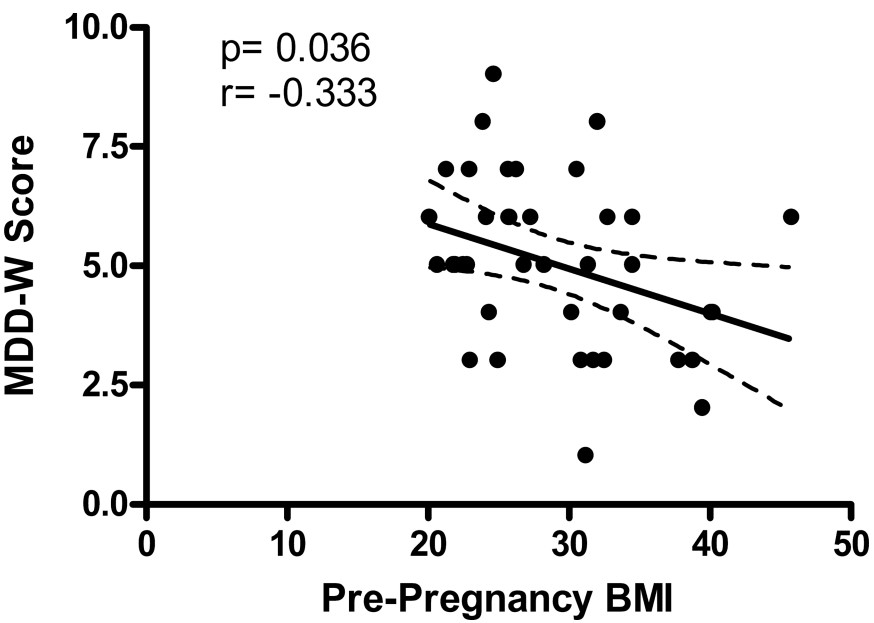

**Figure 1 Pre-pregnancy BMI and MDD-W score during the third trimester of pregnancy are inversely correlated.**

dietary diversity consumed them. Meat, fish and poultry were more commonly consumed than eggs.

## MDD-W score and BMI

Third trimester MDD-W score and maternal pre-pregnancy BMI were inversely correlated (Fig. 1). Most normal weight women (86%) and overweight women (88%) achieved minimum dietary diversity with an MDD-W score greater than or equal to 5 (Fig. 2). Conversely, only 39% of obese women achieved minimum dietary diversity. Significantly fewer obese women achieved minimum dietary diversity compared to normal weight women ($p = 0.012$) and compared to overweight women ($p = 0.036$). Furthermore, third trimester MDD-W scores differed by pre-pregnancy BMI category, however, no pairwise comparisons were significant (Fig. 3). The median MDD-W total for obese women was under 5, while the median MDD-W for overweight and normal weight women was 5 or above.

## Low nutritional density food consumption

Figure 4 represents the consumption of three low nutrient density food groups including, savory snacks, sweets, and sweetened drinks. Although no comparisons across BMI categories or study cohort were statistically significant, numerically, a higher percentage of BABY GUT participants consumed all three low nutrient density food categories than ARCH GUT participants. Three-quarters of all participants consumed sweetened drinks. The MDD-W protocol defines sweetened drinks as beverages such as fruit juices, sodas, chocolate drinks, sweet teas, coffee with sugar, and energy drinks. Overall, consumption of foods with low nutritional density was high in the study populations.
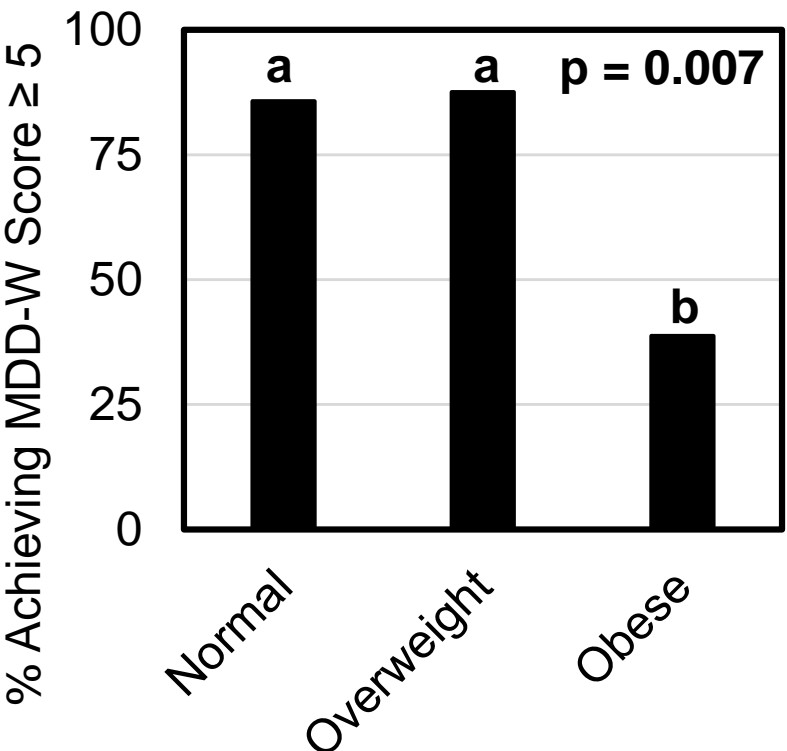

**Figure 2** **A smaller percentage of obese women received MDD-W scores of ≥5 than did normal weight ($p = 0.012$) or overweight women ($p = 0.036$).** The percent of participants achieving a total MDD-W score ≥5 differed by BMI category with 85.7% of normal weight women, 87.5% of overweight women, and 38.9% of obese women attaining MDD-W scores ≥5 ($p = 0.007$). Bars without a common superscript letter differ, $p \leq 0.05$.

## DISCUSSION

In this study, pre-pregnancy BMI was significantly inversely associated with dietary diversity during the third trimester of pregnancy. Fewer women who were obese prior to becoming pregnant reached minimum dietary diversity. Furthermore, women who failed to meet minimum dietary diversity were less likely to consume fruits and vegetables than women who achieved minimum dietary diversity. Low nutrient density foods were consumed frequently by all participating pregnant women.

The result that dietary diversity was inversely correlated with BMI in these pregnant women is similar to results obtained in other studies of non-pregnant women of a similar age. In a study of females 18–28 years of age, it was found that there was an inverse association between dietary diversity score (calculated based on intake of five main food groups and 25 sub-groups) and obesity (*Azadbakht & Esmaillzadeh, 2011*).

Dietary assessment based on dietary patterns is common in nutritional epidemiology (*Hu, 2002*). In fact, various methods of assessing dietary patterns result in common conclusions from different data sets (*Liese et al., 2015*). However, the MDD-W score, a validated method of assessing dietary intake at the population level, minimizes participant

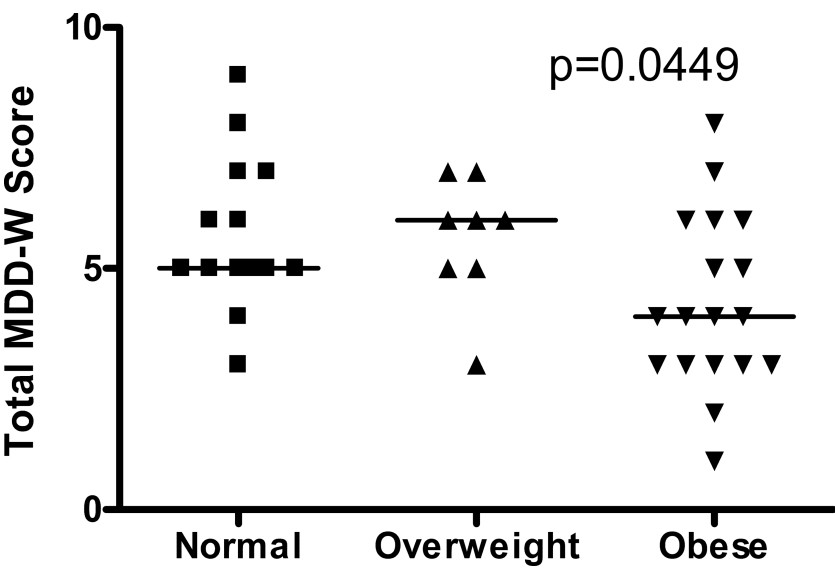

**Figure 3** **Third trimester MDD-W scores differed across the three pre-pregnancy BMI categories with no pairwise comparisons being significant.** The median MDD-W score is indicated by a horizontal line. Parisons being significant.

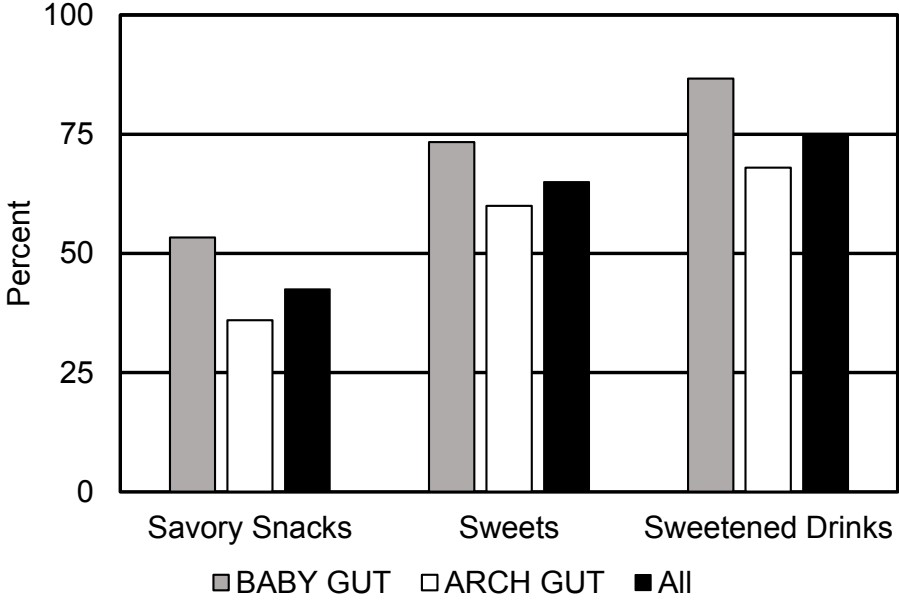

**Figure 4** **Percent of pregnant women consuming specific food groups with low nutrient densities during the third trimester of pregnancy.** Sweetened drinks were consumed by 75% of pregnant women. Intake of these food groups did not differ by BMI category or study population.

and researcher burden. Furthermore, the results from the current study mirror those in other studies focused on dietary patterns. For instance, all three dietary patterns that emerged in a study of obese pregnant women living in the United Kingdom were characterized by consumption of poor diets (*Flynn et al., 2015*). Using methods that did not rely on a measure of dietary diversity, *Mohd-Shukri et al. (2015)* found that obese, pregnant women tended to have a diet that was energy-rich but low in micronutrient density. In four of five low and middle income countries (Bangladesh, Mali, Mozambique, Burkina Faso, and the Philippines) few women achieved dietary diversity, and the prevalence of nutrient adequacy was below 50 percent in pregnant women and even lower among lactating women (*Arimond et al., 2010*). Studies using the Healthy Eating Index have also found an association between higher pre-pregnancy BMI and poor dietary intake among pregnant women (*Laraia, Bodnar & Siega-Riz, 2007*; *Shapiro et al., 2016*; *Shin, Lee & Song, 2016*; *Tsigga et al., 2011*) as has other research assessing dietary quality in pregnancy. The dietary data collected in this study could not be used to calculate the Healthy Eating Index because there was no information about the amount of food consumed.

Over 60 percent of the pregnant women enrolled in the ARCH GUT and BABY GUT studies achieved minimum dietary diversity. This is in stark contrast to similar research conducted in low- and middle-income countries. In Kenya, only 15 percent of the pregnant women, ages 16 to 49 years, achieved dietary diversity (*Kiboi, Kimiywe & Chege, 2017*). The Kenyan study used a slightly more specific diversity score that included 14 food groups instead of 10. However, the FAO had developed the 14-group score and participant scores were based on a 24-hour recall making this study similar to those using the MDD-W. In a systematic assessment of 187 countries, 54 percent of women achieved dietary diversity. MDD-W was not used in that study, however, a similar diet scale was used based on consumption of ten food groups ascertained from a 24-hour recall (*Imamura et al., 2015*). Combined, these results support the importance of considering the geographic location when interpreting results from similar studies and suggest that grouping large populations of women from different geographical areas into a single statistic may be misleading.

Lack of dietary diversity is more commonly considered an issue for women living in low and middle income nations (*Chakona & Shackleton, 2017*). Our research demonstrates that a lack of dietary diversity is also an issue in the United States. One reason cited for lack of dietary diversity is a lack of access to diverse food sources (*Imamura et al., 2015*). In our study, food access was not measured so it's not clear if a lack of access to a variety of foods contributed to the low MDD-W scores in obese women. Other research has demonstrated that individuals living in areas with poor access to healthy foods are more likely to be obese (*Chi et al., 2013*; *Vargas, Stines & Granado, 2017*), so there is a possibility that obesity is a proxy measure for living in a food desert. Providing pregnant women with a door-step delivery of a diverse array of foods is one potential way to improve dietary diversity. This would combat common barriers to intake including poor availability of foods locally, challenges in shopping for healthy foods, time limitations of preparing healthy foods, and lack of knowledge or motivation to buy foods that are unfamiliar. The current study did not collect data to ascertain participants' proximity to a food desert or access to food. Future studies should account for this possibility.

In our study, regardless of pre-pregnancy BMI status, fruits and vegetables were the foods most commonly missing from the diets of women with low MDD-W scores. However, low-nutrient foods were commonly consumed. It could be that low-nutrient density foods are displacing fruits and vegetables from the diets of our participants. This is consistent with other reports of dietary intake among adults in the US (*DHHS & USDA, 2015*). Furthermore, children born to women who consumed the most fruits during pregnancy performed better on cognitive tests at 1 year of age (*Bolduc et al., 2016*) and a meta-analysis concluded maternal diet quality during pregnancy is positively associated with child neurodevelopment and cognitive development (*Borge et al., 2017*). Others report that maternal fruit and vegetable consumption increases infant head circumference (*Loy et al., 2011*). Given these associations and that fruit and vegetable consumption has been associated with numerous health benefits, this low intake of fruits and vegetables by pregnant women is concerning.

As briefly mentioned above, a large proportion of the pregnant women in this study consumed low nutrient density foods. It was surprising that this intake did not differ by pre-pregnancy BMI category, but this is consistent with other reports in the US (*DHHS & USDA, 2015*). For instance, three-quarters of women in ARCH GUT and BABY GUT reported consuming at least one sweetened drink in their 24-hour recall. This is consistent with the 2011–2014 NHANES report that 68 percent of adults living in the Northeast United States consumed at least one sweetened drink each day and 40 percent of all women in the US consume sweetened beverages daily (*Park, McGuire & Galuska, 2015*). These results are important as sweetened beverage consumption is a significant contributing factor to the rise in overweight and obesity as well as associated with a higher risk of hypertension, congenital heart disease, preterm delivery, and gestational diabetes (*Englund-Ogge et al., 2012*; *Malik, Schulze & Hu, 2006*; *Marquez, 2012*; *Xi et al., 2015*). Despite consistency in intakes of low nutrient density foods across pre-pregnancy BMI categories, risk for diseases associated with such intake are only reported for obese women. It is likely that, although there is intake of these foods by all women, women in some BMI categories consume larger volumes of such foods. Unfortunately, the methods used in this research did not assess amounts of food consumed.

There were additional limitations of this research. The study included a modest sample size ($n = 40$) and relied on a single self-reported 24-hour dietary recall. A single recall is not representative of long-term dietary intake. However, the MDD-W is intended to be used with single dietary recalls as it is applicable for use with groups of individuals (in this instance: groups of normal weight, overweight and obese women) (*FAO & FHI 360 , 2016*). The MDD-W is not meant to be used to describe characteristics of individual women. No information about dietary supplement intake or prenatal vitamin consumption was collected from participants. The study population had a higher rate of obesity (45%) compared to those of the state of Michigan (32.5%, all adults (*Centers for Disease Control and Prevention, 2018*) and United States (25%, women prior to pregnancy (*Centers for Disease Control and Prevention, 2018*). The cause of this discrepancy is unknown, but may be related to the method of advertising for the studies. Study recruitment flyers stated, "You can help us learn if there are ways to prevent childhood tummy problems and

obesity." Therefore, women who were obese might have been more likely to participate. Furthermore, this sample is a collection of participants recruited into two separate cohorts. However, all data used in this analysis was collected using the same survey instruments and was collated by the same research staff. Finally, the study was conducted within a geographically limited (one urban and one rural area in Michigan) population.

Future research should analyze the associations between dietary diversity during pregnancy and pre-pregnancy BMI in a larger, more representative study population to confirm the inverse relationship between dietary diversity and pre-pregnancy BMI holds in a number of geographical locations across the United States. In addition, future research could explore these connections stratified not only by geographical location but also by access to food, socioeconomic status or parental education level.

## CONCLUSIONS

Based on a single 24-hour dietary recall, women who were obese prior to becoming pregnant failed to consume a diverse diet during the third trimester of pregnancy. Furthermore, pregnant women who failed to achieve minimum dietary diversity (M-DDW <5) in their third trimester of pregnancy were less likely to consume fruits and vegetables than women who achieved minimum dietary diversity. Diverse dietary intake not only ensures adequate vitamin and mineral intake but also increases fiber and phytochemical intake. Despite being important for human health and a healthy pregnancy, neither fiber nor phytochemicals are found in most prenatal vitamins. Thus, prenatal care providers should emphasize the necessity of consuming a diverse diet when providing nutrition information to pregnant women. Given the results of this research, this recommendation is especially relevant for pregnant women who were obese prior to becoming pregnant, a population that has been demonstrated to be open to behavior change with respect to pregnancy (*Dodd et al., 2014*).

### List of Abbreviations

| | |
|---|---|
| **BMI** | Body mass index |
| **FAO** | Food and Agriculture Organization |
| **MDD-W** | Minimum Dietary Diversity for Women of Reproductive Age |
| **SAS** | Statistical Analysis Software |

## ACKNOWLEDGEMENTS

We thank the ARCH investigators and Comstock Lab members who were not directly involved in the preparation of this manuscript but without whom this project could not have been completed.

### Funding

This research was financially supported by the Rachel A. Schemmel Undergraduate Research Scholarship, and Michigan State University College of Natural Resources Undergraduate Research Program Scholarship. The funders had no role in study design, data collection and analysis, decision to publish, or preparation of the manuscript.

### Grant Disclosures

The following grant information was disclosed by the authors:
Rachel A. Schemmel Undergraduate Research Scholarship.
Michigan State University College of Natural Resources Undergraduate Research Program Scholarship.

### Competing Interests

The authors declare there are no competing interests.

### Author Contributions

- Breanna M. Kornatowski conceived and designed the experiments, performed the experiments, analyzed the data, prepared figures and/or tables, authored or reviewed drafts of the paper, approved the final draft.
- Sarah S. Comstock conceived and designed the experiments, performed the experiments, analyzed the data, contributed reagents/materials/analysis tools, prepared figures and/or tables, authored or reviewed drafts of the paper, approved the final draft.

### Human Ethics

The following information was supplied relating to ethical approvals (i.e., approving body and any reference numbers):

Study procedures for ARCH (IRB#C07-1201), ARCH GUT (IRB# 14-170M) and BABY GUT (IRB# 15-1240) were approved by the Michigan State University IRB.

### Data Availability

The raw data and code are provided in Supplemental Files.

### Supplemental Information

Supplemental information for this article can be found online at http://dx.doi.org/10.7717/peerj.5526#supplemental-information.

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
