# Peer review of "Dietary diversity is inversely correlated with pre-pregnancy body mass index among women in a Michigan pregnancy cohort"

_PeerJ, doi:10.7717/peerj.5526_

## Round 0.1 · original submission · Major Revisions

Dear Dr. Comstock,

Your paper has been seen by the editor and three external expert referees. The reviewers found the paper to be clear and the findings to be of interest. However, they raised some concerns which need to be considered. If these can be satisfactorily addressed, a revised manuscript is likely to be suitable for publication. I enclose below the comments received that set out a number of points which will need your attention before we can consider the submission further. I would urge you to give these points your careful attention; in particular, the concerns raised regarding the focus of the paper and the single use of the dietary recall, which is the main and major limitation of the study.

Regards,

Stefano Menini

Reviewer 1 ·

Basic reporting

Please see point 4.

Experimental design

Please see point 4.

Validity of the findings

Please see point 4.

Additional comments

GENERAL COMMENT
The authors aimed to evaluate the association between third trimester dietary diversity and self-reported pre-pregnancy BMI in a sample of 100 pregnant women. The authors found that Women who were obese prior to becoming pregnant failed to consume a diverse diet. Furthermore, women who failed to achieve minimum dietary diversity (M-DDW < 5) were less likely to consume fruits and vegetables than women who achieved minimum dietary diversity. This paper is interesting, however, concerns are raised regarding the focus of the paper, result presentation and discussion.


SPECIFIC COMMENTS

1- The major concern of that study is the single use of the dietary recall. This should be deeply acknowledge in the discussion section and in the conclusion. The authors should be careful not to suggest that this is an evaluation of dietary diversity during pregnancy, because this represents the third trimester diet.
2- The abstract is far too long; the background should be shortened to 2 or 3 sentences. The objective should be clearly specified as well as the sample size.
3- This sample of US pregnant women is composed of a large proportion of obese women (45%), is that representative? If not, please acknowledge and explain why in the discussion section.
4- The authors emphasize on micronutrient supplementation both in the introduction and in the discussion, while they are not measuring it. Do they have this information ?
5- In the introduction section, please clearly specify the objective AND the hypothesis.
6- In the methods section, I would not use the term small for the studies. Just information on sample sizes is sufficient.
7- In the methods section, please give more information on the diet recall assessment. Was it self-administered ? Was it verified by a dietician ?
8- Was researchers reviewing the scores were registered dietician ?
9- What about the MDD-W in vegetarians or vegans and its association with health issues ? This could be discussed.
10- I would like to read more about the choice of using the MDD-W score instead of the HEI. Not only one study was performed during pregnancy using HEI…
11- The discussion lacks of focus. This section should be oriented around authors’ findings.
12- In the discussion section, one of the most interesting findings is not discussed: women who failed to achieve minimum dietary diversity (M-DDW < 5) were less likely to consume fruits and vegetables than women who achieved minimum dietary diversity.
13- Why the authors are discussing about food desert if they did not evaluate those concerns ?
14- Why are the authors are talking about microbiota at the end, while this subject is not discussed throughout the MS ?
15- There are too many figures. They are all not necessary.



MINOR COMMENT :
1- Please do not use abbreviation for women of reproductive age.
2- Please revise the sentence on line 120.
3- Why are the limitations section is presented before the discussion?

·

Basic reporting

This is a really nicely written article. English is perfect and the text is easy to follow and understand. There is a correct use of references, is well discussed even if not extent of limitations (as the authors reported only one 24HDR is clearly not representative enough) Still there are a couple of issues that remain, to me, a bit unconnected.

The first refers to the intake of Low Nutritional Density Food Consumption; while the introduction is mainly focused on the dietary diversity score you come across suddenly with the results on Low Nutritional Density Food Consumption. I think this point also requires to be introduced, for example, what is a low nutritional density food? how is it defined? what is the importance of check in on these also in this study?

Another point that remains a bit unconnected to me is the very last part of the discussion (line 324-328), where you talk about the gut microbial community. To me there are hundreds of other topics related with your work that could be mentioned, microbiome is only one, and placing this discussion there gives it some importance that to me is too much.

Minor comments:
Table 1: I am bit lost with the p values, a bit more detailed footnote could be helpful. For example, I do not know what are the test for BMI (p=0.1395); Age Group (p=0.5766); Parity (p=0.1030)
Table 2: The asterisk, has any statistical test been applied there?
Figure 2: What are "a" and "b"?

Experimental design

No comment

Validity of the findings

As I did state before and you have already noted in your limitation, the inclusion of only one 24HDR is limited in assessing diet. This is clear, but you have been able to put your results in the context of other studies.

Reviewer 3 ·

Basic reporting

1- There is not enough literature to show that dietary diversity is a predictor of obesity.
2- Too many information has been given in the introduction which is not required and should be removed or put in the discussion.
3- The background of the study was not explained in the introduction. Previous literature in Americans or other parts of the world should be stated.

Experimental design

1- One 24-hour dietary recall cannot show the regular eating pattern. Food frequency questionnaire or either three 24-hour dietary recall may be a better choice.
2- Measuring dietary intakes during pregnancy may not a good indicator to investigate its association with pre-pregnancy BMI. It is not logical.

Validity of the findings

1- No analysis has been given for Table 2. The low sample size (n=40) has restricted to conduct for further analysis. (low power)
2- Table 1 is not informative.
3- Findings were more descriptive rather than analytical.

Additional comments

By measuring dietary intakes during pregnancy, conclusion cannot be made.

---

## Round 0.2 · accepted · Accept

Dear Dr. Comstock,

Our referees have now considered your paper and have recommended publication in “PeerJ”. We are pleased to accept your paper in its current form which will now be forwarded to the publisher for copy editing and typesetting.

I thank all reviewers for their effort in improving the manuscript and the authors for their cooperation throughout the review process

Yours sincerely,

Stefano Menini

Reviewer 3 ·

Basic reporting

Authors have fulfilled the reviewers' comments on the Title of MS, introduction, and relevant citations.

Experimental design

The objective and hypothesis of the research is now addressed clearly.

Validity of the findings

no comment

Additional comments

The manuscript has been improved significantly in particular in the limitation section.